# A Pilot Study Examining the Effects of Music Training on Attention in Children with Fetal Alcohol Spectrum Disorders (FASD)

**DOI:** 10.3390/s22155642

**Published:** 2022-07-28

**Authors:** Dathan C. Gleichmann, John F. L. Pinner, Christopher Garcia, Jaynie H. Hakeem, Piyadasa Kodituwakku, Julia M. Stephen

**Affiliations:** 1The Mind Research Network, 1101 Yale Blvd NE, Albuquerque, NM 87106, USA; dgleichmann@mrn.org; 2Department of Psychology, University of New Mexico, 2300 Menaul Boulevard NE, Albuquerque, NM 87107, USA; pinnerj@unm.edu (J.F.L.P.); cmgarcia024@gmail.com (C.G.); 3Music Therapy—A Sound Approach, 1212 Daskalos Dr. NE, Albuquerque, NM 87123, USA; hakeems@comcast.net; 4Center for Development and Disability, Department of Pediatrics, University of New Mexico, 2300 Menaul Boulevard NE, Albuquerque, NM 87107, USA; kodpw@unm.edu

**Keywords:** EEG, attention networks test, fetal alcohol spectrum disorders, music training, prenatal alcohol exposure, neurodevelopment

## Abstract

Prior studies indicate differences in brain volume and neurophysiological responses of musicians relative to non-musicians. These differences are observed in the sensory, motor, parietal, and frontal cortex. Children with a fetal alcohol spectrum disorder (FASD) experience deficits in auditory, motor, and executive function domains. Therefore, we hypothesized that short-term music training in children with an FASD due to prenatal alcohol exposure may improve brain function. Children (*N* = 20) with an FASD were randomized to participate in either five weeks of piano training or to a control group. Selective attention was evaluated approximately seven weeks apart (pre-/post-music training or control intervention), examining longitudinal effects using the Attention Networks Test (ANT), a well-established paradigm designed to evaluate attention and inhibitory control, while recording EEG. There was a significant group by pre-/post-intervention interaction for the P250 ms peak of the event-related potential and for theta (4–7 Hz) power in the 100–300 ms time window in response to the congruent condition when the flanking stimuli were oriented congruently with the central target stimulus in fronto-central midline channels from Cz to Fz. A trend for improved reaction time at the second assessment was observed for the music trained group only. These results support the hypothesis that music training changes the neural indices of attention as assessed by the ANT in children with an FASD. This study should be extended to evaluate the effects of music training relative to a more closely matched active control and determine whether additional improvements emerge with longer term music training.

## 1. Introduction

The prevalence of fetal alcohol spectrum disorders (FASD) ranges from 1–5% in the US [1]. Despite this high prevalence, few evidence-based interventions exist for children with an FASD and most of these target behavior rather than cognitive function [2]. Children with an FASD experience a broad range of deficits involving auditory processing [3], motor sequencing [4], and executive functions, such as attention and working memory [5,6]. These results support the need for interventions targeting sensorimotor and executive function deficits to improve long-term outcomes for individuals with an FASD. These deficits are accompanied by a wide range of alterations in brain function and structure including a reduced size of the corpus callosum, impaired sensory processing and alterations in fronto-parietal networks [3,7,8,9,10]. While these deficits have been well-documented, additional studies [11,12] provide evidence for neural plasticity following prenatal alcohol exposure (PAE), including results from animal studies indicating that enriched environments can alleviate intermediate changes in brain structure/function due to PAE.

Considerable evidence supports the ability of music training (MT) to induce neuroplasticity in healthy controls with retrospective studies of adults revealing structural and functional differences in musicians versus non-musicians in motor [13] and auditory cortices [14], as well as higher-order regions [15,16]. It is also posited that neuroplasticity occurs with MT across the lifespan including in childhood. In a large study of structural brain development, Hudziak et al. [13] reported a faster rate of developmental change in cortical volume in children with MT in a broad network including motor, premotor, supplementary motor, parietal, and prefrontal cortices. In a prospective study, Kraus et al. [17] identified improved language learning and altered neural signatures of speech encoding in healthy children with two years of MT. Other prospective studies also report differences in behavioral measures such as attention and working memory [18,19], adding evidence for the role of MT in changing brain structure and function in otherwise healthy individuals.

The strong evidence for differences in brain structure and function between musicians and nonmusicians has led research groups to examine the possibility of using MT to improve brain function in individuals with brain-related disorders [20,21,22,23,24]. Retrospective studies are limited in their ability to determine causality of effects, and one possible explanation for these retrospective results is that musicians have a unique aptitude for music that explains these differences in brain structure and function. However, Barrett et al. [25] argued that prior studies demonstrating both dose effects (amount of MT) and instrument-specific effects (e.g., increased lateralized hand representations within the brain [26] of string players) indicate that MT is likely causing brain changes in these retrospective studies rather than a simple predisposition for music. Additional prospective studies have provided evidence for neuroplasticity with music training across the lifespan. For example, in healthy individuals aged 64–76 years of age, brain volume increased in five music-related brain regions in those with piano practice vs. controls [27]. In younger individuals, structural changes in the auditory cortex are evident in 9–11 year old children with MT vs. nonmusicians [28]. These results provide support for the role of MT in inducing neuroplasticity in individuals with alterations in brain function and structure across the age-span.

Attention has been studied widely using neuroimaging with Posner identifying three unique but overlapping attention networks within the brain: the alerting system, the orienting system, and the executive attention network [29]. Subsequent work in the field of neuroimaging has contributed new knowledge regarding the brain regions involved in mediating attention, including fronto-parietal networks for selective attention, anterior cingulate cortex to resolve conflicting stimuli, and temporoparietal junction for disengagement of attention [30]. The Flanker task was designed specifically to elucidate the different attention networks within the context of a visual selective attention task. Selective attention is a critical skill throughout development [31], allowing a child to focus on the task at hand (e.g., listening to the teacher) without being distracted by other stimuli (e.g., classmates whispering). The child version of the Flanker task (the Attention Networks Test—ANT) incorporates all of the components of the adult version in the context of a game and has been validated in children as young as 4 years of age [32,33]. Performance of the ANT generates an event related potential eliciting a peak occurring fronto-centrally between 200–300 ms poststimulus [34]. Furthermore, fronto-central theta oscillations have been reliably reported in tasks with conflicting stimuli [35] including the ANT [36]. McDermott et al. [37] demonstrated that the ANT is valid for children as young as 4–6 years of age.

EEG, a widely-available neurophysiological approach, is well-suited to evaluate the effects of MT on brain function due to its direct measure of neural activity combined with high temporal resolution [38]. The current study examined the effects of MT in children with an FASD by measuring EEG during performance of the ANT pre- and post-intervention [34,39]. We hypothesized that MT would improve attention through activation of brain regions involved in executive control. These changes would occur through improved neural synchrony, with amplitude directly related to the number of neurons synchronously active at any instant in time within the millisecond timeframe and evidenced by increased amplitude of EEG components (e.g., N250 response) or increased spectral power of neural oscillations (e.g., theta) in children with MT relative to controls.

## 2. Materials and Methods

### 2.1. Participants

The study was approved by the University of New Mexico Health Sciences Center Human Research Review Committee in accordance with the Declaration of Helsinki. A total of 20 children (12 boys, 8 girls) with a fetal alcohol spectrum disorder (FASD), ages 5 to 10 years (Mean = 7.58 years) were recruited through the Neurodiagnostic Clinic at the Center for Development and Disability, University of New Mexico School of Medicine. Children diagnosed with fetal alcohol syndrome (FAS) had a confirmed history of prenatal alcohol exposure and demonstrated the characteristic constellation of dysmorphic features, including growth restrictions (height and weight less than 10th percentile), facial anomalies (e.g., short palpebral fissure, abnormalities in the premaxillary zone), and evidence of neurodevelopmental abnormalities. Those diagnosed with alcohol related neurodevelopmental disorder (ARND) also had a confirmed history of alcohol exposure but did not have the foregoing constellation of dysmorphic features. On formal neuropsychological assessment the individuals with ARND were found to have cognitive difficulties including intellectual impairment and deficits in specific domains of functioning (e.g., executive function). The FASD diagnoses followed the revised Institute of Medicine criteria including the criteria for documented prenatal alcohol exposure [40].

Informed consent was obtained from all subjects involved in the study. Informed consent was obtained from the parents and assent was obtained for children 7 years and older. Families were interviewed to verify inclusion (e.g., age and feasibility of a time commitment for eight consecutive weeks) and exclusion criteria (e.g., history of major psychiatric or neurological problems or sensory-motor impairments). None of the participants had received prior lessons in piano or other musical instruments. The children were randomly assigned to the MT or control group via a random number selection. In one case, a sibling was re-assigned to the MT group so that both siblings were engaged in the same study arm to reduce potential confounds. Privacy was protected by utilizing anonymized study IDs. The racial/ethnic composition of the participants reflected the demographics of the clinic from which the participants were recruited and included 7 American Indian—non-Hispanic, 8 Hispanic, 3 non-Hispanic White, and 2 unreported. Race and ethnicity were approximately evenly distributed between MT and control groups (e.g., 4 vs. 3 American Indian) but was not evaluated statistically due to the small sample size.

### 2.2. Cognitive Ability and Musical Aptitude

The Kaufman Brief Intelligence Test, Second Edition (KBIT-2 [41]) was used to assess the participants’ general cognitive ability. The KBIT-2 consists of three subtests, two that estimate verbal intelligence and one that estimates nonverbal intelligence and can be administered by psychologists and non-psychologists. The Primary Measures of Music Audiation [42] were used to assess music aptitude and measure the ability to hold two short musical phrases in working memory and judge their similarity for rhythm or tonal patterns.

### 2.3. Music Training

Since the data reported here were collected in 2013, this study was not registered within clinicaltrials.gov based on the lack of precedent for behavioral trial registration at the time of study completion.

A certified music therapist provided musical keyboard training in 30-min weekly lessons over five consecutive weeks. The first lesson consisted of introducing the keys on the keyboard with colored stickers. Each color corresponded to a musical note and the finger and key that was to play the note. If necessary, the colored stickers were placed on the participants’ fingers to remind the participant which finger played which key. The music training sessions were hierarchically structured, with lessons progressing from unimanual to bimanual training. Melodic line, rhythmic components, and pitch discrimination were also introduced as the sessions progressed. To evaluate musical understanding, the participants were asked to compose a piece including musical notes and/or musical concepts introduced during the session. After each lesson, the participants were given musical excerpts and their compositions to practice. The participants took an electric keyboard home and were expected to practice 10 min a day with the understanding that they would be performing their musical composition the following week. Participants kept a log of daily practice and the clinical psychologist met with the family to review the log every week. Seventeen participants (85%) completed the program including pre/post EEG assessments.

### 2.4. Control Condition

A clinical psychologist met with the families of the control group weekly to discuss any concerns about their children’s behavior and emotional functioning. The children in the control group engaged in free play with a research assistant in a separate room while the parent(s) met with the clinical psychologist.

### 2.5. Attention Network Test (ANT)—Child Version

The child version of the ANT (see Figure 1) was employed to assess attention [32]. For each trial, the child was presented with a visual fixation cross on a blue background screen. Following a variable delay (400–1600 ms), one of four cue (duration 150 ms) conditions indicating where to attend was presented (1. No cue, 2. Spatial cue—attend either above or below fixation, 3. Double cue—attend above and below fixation, or 4. Attend centrally). Following a 400 ms delay, target stimuli were presented, which consisted of a line drawing of five yellow fish with the four flanking fish facing either all to the left or all to the right. The central fish also faced either to the left or right, but an equal number of trials were presented, where the central fish was congruent with the flanking fish or incongruent with the flanking fish. The task was to decide whether the middle fish was pointing to the left or right with a button press. The children were instructed to “feed the fish” by deciding what direction the middle fish was pointing (left or right). The children received feedback (video of the fish chewing with bubbles emerging for correct or beep for incorrect). An additional condition presented the central fish without any flanking fish (solo). Fish were presented above or below fixation to the attended or unattended location. Task duration was ~15 min.

The effects of MT were examined within the three attention networks of the ANT through examination of the alerting, orienting and the conflict reaction times (RTs). These RTs are defined in Fan et al. [43] as: alerting RT = no cue − double cue, orient RT = center cue − spatial cue, and conflict RT = incongruent − congruent.

### 2.6. EEG

EEG data were collected pre-/post- the 5-week intervention at an average time between EEG assessments of 7.3 weeks. One participant in the music group received their post-MT assessment 4 months after the initial EEG due to family scheduling conflicts. Other than this outlier, pre-/post-assessment timing was similar across groups.

EEG data were collected using a high-density active electrode 124-channel Biosemi system. The Biosemi EEG was housed in a shielded and sound-proofed room. Data were collected at 1000 Hz with a 0.01–300 Hz online filter. Offline EEG data analysis was performed using EEGLAB. The data were low-pass filtered at 50 Hz and epoched from −300–1800 ms relative to target onset for congruent, incongruent, and solo conditions. In addition to removing high frequency noise, the 50 Hz low-pass filter effectively eliminated noise due to 60 Hz line noise. Baseline correction was applied. Bad channels and epochs were rejected based on visual inspection and outlier epochs (amplitude >1000 µV peak-to-peak rejected automatically). After identification and imputation of bad channels, an average reference was applied. Two accessory channels located above and below one eye were used to identify and eliminate eye blink artifacts via ICA. Once preprocessing was complete, the retained epochs for each condition (congruent, incongruent, solo) were averaged. The resulting event related potentials (ERPs) were analyzed by identifying the prominent peaks in the waveform, extracting the peak amplitude and latency using ERPLab, and performing statistical analysis of the resulting data. To reduce multiple comparisons, responses were averaged across four fronto-central midline channels extending from Cz to Fz to capture the fronto-central response reported previously. A 15 Hz low pass filter was applied to the ERP data prior to identification of peak amplitude and latency to reduce the influence of noise on these values. One prominent peak (200–300 ms) was identified in the ERP and amplitude and latency values were extracted from this peak using the peak amplitude and peak latency measurement extraction tools in ERPlab.

To examine theta oscillations, we calculated the event-related spectral perturbation (ERSP) in EEGLAB [44] using the default wavelet parameters of 3 cycles and a scaling factor of 0.5, which slowly increases the number of cycles as the frequency increases, reaching 50% more cycles for the highest frequency of 110 Hz, for the congruent and incongruent conditions for the retained epochs described above. The time window for analysis was (−500, 500) ms relative to stimulus onset. The prestimulus baseline time interval (−500, 0) was used to calculate the ERSP, which identifies increases/decreases in spectral power relative to the baseline time interval. Based on the role of frontal theta oscillations in executive function [35], we examined theta power in frontal/central midline channels similarly to what is described above for the ERP analysis. To examine stimulus specific changes in spectral power and to avoid edge-effects, the average ERSP value from fronto-central channels was extracted from the time/frequency window of 100–300 ms and 4–7 Hz for each participant and analyzed statistically.

### 2.7. Statistical Analysis

After extracting the latency, amplitude, and spectral power values, the EEG data were analyzed using a repeated measures analysis of variance with pre-/post intervention EEGs as a within subjects factor and a group (MT vs. control) as the between subjects factor. To reduce the number of multiple comparisons, the EEG signal in fronto-central midline channels from Cz to Fz was analyzed similar to Rueda et al. [33] and Monastra et al. [45]. Statistical analysis was performed using SPSS v. 20 (IBM, Armonk, NY, USA).

## 3. Results

Table 1 presents the participants’ demographic characteristics. Participants (*N* = 17), for which all pre-/post- measures were available, were well-matched on age (*p* = 0.72) and gender (χ^2^ = 0.03, *p* = 0.85) between the MT and control groups (Table 1). The groups were also well-matched on KBIT scores and tonal and rhythm abilities (*p*’s > 0.05) prior to training. There were no significant differences in the number of retained trials either between groups or between time points within group (*p*’s > 0.1). The same trials were incorporated in the ERP and the ERSP analysis. There was a trend for a significant (*p* = 0.051, partial η^2^ = 0.25) group x time point interaction for the Hit RTs across all conditions of the ANT indicating a trend for improvement in RT following training in the MT group only. There were no significant pre-/post-, group or pre-/post- × group interactions for the ANT attention RTs (Table 2; *p*’s > 0.12).

Good quality pre-/post intervention EEG data were obtained for 17 participants. Figure 2A shows the pre-/post-training, group-averaged ERP for the congruent condition of the ANT. Amplitudes and latencies for the 200–300 ms peak are shown in Table 3. No significant group differences or group by time point interactions were obtained for the incongruent condition. Positive peak amplitude in the 200–300 ms time frame demonstrated a significant training group by time (pre vs. post) interaction for amplitude for the solo and congruent conditions and for latency for the solo condition only (Table 4). As shown in Figure 2A, this P250 peak was equivalent between groups pre-training, but was different post-training, as shown in Figure 2B. The significant group by time (pre- vs. post-intervention) interaction for theta oscillations demonstrated increased theta power during performance of the ANT following training in the MT group only (F(1,13) = 6.2; *p* = 0.026, partial η^2^ = 0.325) (Figure 2C). There was no significant group by time interaction for the incongruent condition (*p* = 0.9).

## 4. Discussion

This preliminary investigation in children with an FASD provides evidence for neurophysiological changes following MT. Importantly, the children were randomized to group by the study team and the groups were well-matched on age and intellectual functioning prior to MT. Despite randomization, the MT group had a larger representation of children with an FAS diagnosis than the control group. It is currently unclear whether differences in the FASD subtype will lead to differential effects of MT especially given that the pre-training cognitive assessments were similar across groups. Furthermore, the ERP and ERSP measures were similar between groups in the first EEG session prior to training. At the second EEG session, the children who received MT had higher ERP peak amplitudes and higher theta power compared to the control group. Finally, there was a trend (*p* = 0.051) for faster hit reaction times in MT children compared to controls. The significant group x training effects provide support for the role of MT in the improvement of neural indices of attention, as hypothesized.

Prior studies provide support for the ability of short- and long-term MT to influence brain function and structure. Barbaroux et al. [46] demonstrated that 18 months of classical music training provided to a group of 18 children from low income households improved general intelligence, processing speed, and concentration ability as measured by the WISC-IV pre- vs. post-training. An EEG study by Cheung et al. [47] discovered that between two age-, education-, gender-, and cognitive ability-matched groups, those that underwent a year of MT had improved verbal memory relative to controls and theta coherence during verbal memory encoding was positively associated with verbal memory performance. Our results are also in line with an EEG study by Moreno et al. [48], who also performed short-term MT in children with no music experience and reported a post-training increase with enhanced late discriminative negativity amplitude during a passive auditory oddball paradigm. A longitudinal study by Habibi et al. [49] revealed that children in the MT group had a difference in response to frontal regions during the performance of a visual Stroop task relative to age-matched typically developing controls, but their response did not differ from similarly matched children in sports. These results were in line with Hudziak et al. [13], whose longitudinal structural MRI study over six years revealed that MT is associated with more rapid cortical maturation in the areas of the brain responsible for motor function, impulsivity, and visuospatial ability, and provide supporting evidence for the effects of MT on brain function and attention. However, there are no published studies examining MT in children with an FASD with which to compare the current findings.

Many MT studies in typically developing controls have been implemented within the framework of a long-term music training program, e.g., Barbaroux et al. [46]. However, MT studies for clinical populations have typically utilized shorter duration MT sessions within a music therapy framework [50,51]. There is some indication that clinical populations may reveal larger effect sizes despite shorter intervention duration [50,51]. Future studies are needed to determine the optimal duration of MT for improving outcomes in clinical groups.

It is important to note that the significant group by time point interaction in the amplitude of the P250 peak is due to both a *decrease* in peak amplitude in the control group and an *increase* in amplitude in the MT group. While it is surprising to see a decrease in amplitude in the control group, a similar pattern was observed in the significant interaction for the solo condition, providing further support that this does not represent spurious results. Adult studies indicate that test-retest reliability is high for the ANT, when assessing the responses within the same session [52]. On the other hand, test-retest reliability of the ANT in children is not as well established with one small study [32] revealing a lack of correlation in neurophysiological responses across a 6 month timeframe, but a reasonable correlation of even/odd averages was observed within a single session. Due to the limited literature on test-retest effects in the ANT in children, it is difficult to determine whether the decrease in amplitude in the P250 peak is expected in the control group.

While the ANT is a well-established EEG paradigm for assessing attention, it has not been widely used for within-subject examination of the effects of an intervention [33]. Yet, the P250 peak was significantly different between congruent and incongruent conditions in untrained children, indicating the relevance of this component to visual selective attention [33]. Consistent with Rueda et al. [33] the effects in an attention-trained group were also observed in the 200–300 ms time window. Therefore, we attribute the changes in the P250 peak to a neural index of attention. However, associations with behavior are important to establish in subsequent studies. 

Similar to the ERP results, the significant change in theta power was observed in the congruent condition of the ANT. However, prior cross-sectional studies examining differences in attention between groups focused on the incongruent condition or the difference in response between the congruent and incongruent conditions. Interestingly, an intervention study using meditation as the active arm also reported robust theta oscillations during performance of the ANT and they attributed changes in white matter structure to the strong theta oscillations originating from anterior cingulate cortex [53]. This result indicates that theta oscillations are robustly activated in the ANT in typically developing children and may indicate the increase in theta power in the MT group corresponds to a normalized neural response. With little work examining the within-subject effects of an intervention on the performance of the ANT, a replication study is needed to determine which condition is most likely to reveal changes due to MT.

The results provide preliminary evidence for the possible role of MT in improving attention for children with an FASD, who are known to experience attention problems [5,6]. However, a recent meta-analysis [54] concluded that there was no evidence for an effect of MT for the far transfer of skills—or the ability of MT to improve functioning in cognitive skills that are not explicitly trained through MT—in typically developing children. Typically developing children may have limited capacity to improve skills over short periods of time due to performing near or at optimal levels. Prior work indicated greater improvement due to MT in children from low SES households [55], where children from low SES households experience attention deficits relative to children from higher SES households [56,57]. That is, children who have the most to gain from MT may exhibit the largest effects. Children with an FASD experience deficits in multiple domains that are targeted by MT, including auditory processing, sensorimotor function, and attention deficits [3,4,5], which may lead to greater benefit from MT in individuals with an FASD. For example, children with an FASD experience fine motor deficits [58] and thus the fine motor training associated with MT may explain the improvement of RT in the MT group relative to the control group. The possibility remains that non-music related training targeting fine motor or attention function in children with an FASD may also lead to beneficial effects. Future studies in children with an FASD are needed to determine the specificity of effects of MT, to examine whether MT normalizes brain function relative to typically developing children, and to evaluate whether behavioral gains are achieved along with neural changes through an increased duration or intensity of MT.

One of this study’s strengths is that both groups were comprised of children with a diagnosed FASD, which may remove some confounds that exist when examining clinical populations vs. controls, including being well-matched on cognitive performance. However, limiting the sample to children with FASD also limits our ability to determine whether brain function became more similar to that measured in typically developing children following MT. An additional strength of the study was that all children enrolled were naïve to formal MT. Consistent with this, musical aptitude based on tonal and rhythm measures was also matched by the group prior to MT.

The primary limitation of this study was the small sample size. This limitation was offset by the within-subject design carried out over a relatively short time window (7 weeks), limiting the likelihood that other factors explained the group by time point interactions. A second limitation was the control group. The parents of the control group met with a clinical psychologist to discuss behavioral methods to improve outcomes to minimize the differences in study team interaction between groups. Meanwhile, the children of the control group engaged in free play with a trained staff member. Therefore, both groups received weekly interactions with study staff. But the current study cannot conclude whether the attention benefits are specific to music or if a well-designed cognitive training program would lead to similar results. Regardless, a larger study with an active control group would better address this question. Finally, a larger study is needed to examine whether MT might have differential effects across FASD subtype.

## 5. Conclusions

This pilot study provides evidence that children with an FASD may benefit from MT. Similar to other studies, the effect sizes were larger for the neurophysiological measures relative to behavioral measures and may indicate that brain changes precede behavior change or neurophysiological measures may provide increased sensitivity to brain changes relative to behavioral measures. Future longitudinal studies are needed to examine whether continued MT leads to further neurophysiological changes that translate to behavioral change. Additional studies, which include typically developing controls, are also needed to determine whether MT in children with an FASD leads to a normalization of brain function or whether the changes represent a compensatory change.

## Figures and Tables

**Figure 1 sensors-22-05642-f001:**
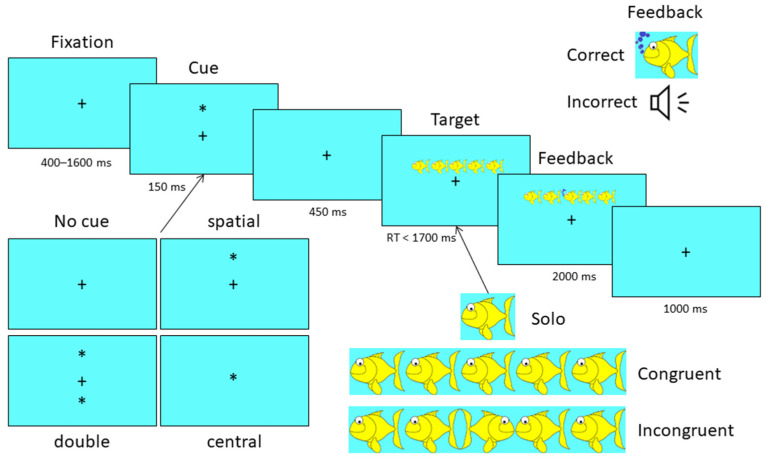
A depiction of the Attention Networks Test (ANT) displaying the four possible cues presented in the lower left corner, the three possible Target conditions of solo, congruent and incongruent in the lower right corner and the correct vs. incorrect feedback conditions presented in the upper right corner. The + denotes the position on the screen at which the child is asked to fixate. The * denotes where the child’s attention should be directed for the Target stimulus. The blue circles denote the fish “eating” as a correct response. The colors of the background and the fish were chosen in the original task to engage children. The sequential boxes demonstrate the timing of the display of each component of the trial. The presentation of the cues and conditions are randomized across trials.

**Figure 2 sensors-22-05642-f002:**
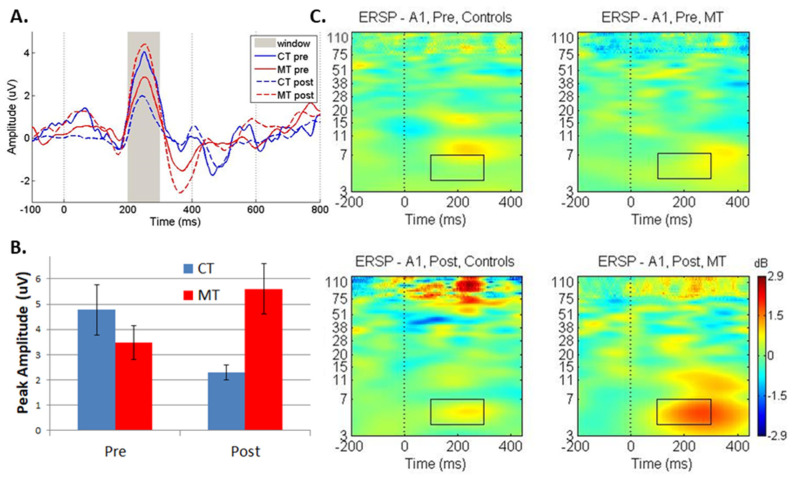
(**A**) The group averaged ERP waveform from the midline fronto-central averaged EEG electrodes for the pre-/post-training EEG sessions for controls (CT) and music trained (MT) participants. The peak latencies and amplitudes were chosen from the 200–300 ms time window (gray shading) using ERPlab. (**B**) The average peak amplitude of the positive peak in the 200–300 ms time window for pre-/post- EEG sessions. Error bars represent standard error of the mean. (**C**) Pre- (**top**) and post- (**bottom**) ERSPs for control (**left**) and MT (**right**) groups are shown with the black box denoting the time frequency window of mean theta power evaluated for group by time point-interactions. The ERSP for the A1 electrode is shown.

**Table 1 sensors-22-05642-t001:** Demographic characteristics.

	Music Training (*N* = 9)	Control (*N* = 8)	*p*-Value
Age (mean, SD)	7.67 (1.80)	8.00 (1.87)	0.72
Sex (Male/Female)	6/3	5/3	0.85
Diagnosis (FAS/pFAS/ARND)	5/0/4	0/2/6	0.20
KBIT Verbal (mean, SD)	88.44 (17.17)	85.00 (11.13)	0.64
KBIT Nonverbal (mean, SD)	92.33 (12.09)	95.40 (8.44)	0.56
KBIT Composite (mean, SD)	88.89 (14.65)	88.00 (9.89)	0.89
Tonal (mean, SD)	29.11 (4.70)	32.80 (1.92)	0.056
Rhythm (mean, SD)	28.55 (4.39)	28.80 (2.77)	0.89
ANT Hits RT Pre (ms; mean, SEM)	1029 (53.1)	1000 (60.3)	0.31
ANT Hits RT Post (ms; mean, SEM)	898 (52.6)	1017 (59.7)	0.0006
ANT #epochs Session1-Congruent	35.3 (2.1)	34.7 (3.3)	0.71
ANT #epochs Session2-Congruent	33.1 (3.2)	31.4 (3.5)	0.36

**Table 2 sensors-22-05642-t002:** ANT Attention RTs (ms): Mean (standard deviation).

Condition	MT-Pre	MT-Post	Control-Pre	Control-Post
Alterting RT	58 (101)	110 (75)	15 (53)	87 (132)
Orienting RT	45 (73)	77 (74)	51 (86)	62 (100)
Conflict RT	139 (99)	146 (76)	199 (112)	116 (91)

**Table 3 sensors-22-05642-t003:** ANT ERP amplitudes and latencies pre-/post-intervention.

Condition	MT-Pre	MT-Post	Control-Pre	Control-Post
Amplitude (μV)				
Solo	3.37 (1.97)	5.21 (2.74)	4.13 (1.77)	1.71 (1.10)
Congruent	3.48 (1.76)	5.60 (2.66)	4.78 (2.66)	2.30 (0.78)
Incongruent	3.88 (2.10)	5.75 (3.72)	4.12 (2.25)	2.77 (1.63)
Latency (ms)				
Solo	244 (27)	261 (29)	252 (24)	238 (18)
Congruent	243 (23)	232 (25)	241 (29)	251 (23)
Incongruent	240 (21)	242 (23)	237 (19)	239 (11)

**Table 4 sensors-22-05642-t004:** ANT ERP statistical analysis results.

Condition	Solo		Congruent		Incongruent	
Amplitude	F-statistic	*p*-value	F-statistic	*p*-value	F-statistic	*p*-value
Pre-Post	0.15	0.71	0.08	0.78	0.08	0.77
Group	3.33	0.093	1.17	0.3	1.68	0.22
Pre-Post × Group	8.11	0.015 *	12.6	0.004 *	3.49	0.086
Latency						
Pre-Post	0.06	0.81	0.004	0.952	0.16	0.70
Group	0.39	0.54	0.61	0.45	0.11	0.75
Pre-Post × Group	8.21	0.014 *	1.35	0.267	0.01	0.92

* Significant at corrected *p*-value of 0.05/3 = 0.016.

## Data Availability

The data presented in this study are available on request from the corresponding author. The data are not publicly available due to data privacy concerns and the lack of approval for data sharing in the original consent form.

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
