# Peer review of "A Pilot Study Examining the Effects of Music Training on Attention in Children with Fetal Alcohol Spectrum Disorders (FASD)"

_sensors, 2022, doi:10.3390/s22155642_

Round 1
Reviewer 1 Report
This paper addresses the possibility that musical training may alleviate some of the deficits in attention encountered by children with fetal alcohol exposure. The study methodology was sound, though impact suffers from very low sample size and a somewhat unclear definition of the extent of fetal alcohol exposure symptoms within the sample. In addition I also have some questions and concerns related to the EEG/ERP analyses. I think the introduction is somewhat sparse, especially when it comes to explaining the potential importance of music training to the difficulties encountered by children with FAS. Some of the text in the discussion could perhaps be moved into the introduction in order to ground the reader. Some of the acronyms for FAS symptoms are not defined--for example, ARND (Alcohol related neurodevelopmental disorder) is included in the table of demographics but not defined anywhere. A more comprehensive explanation of what these different types of FAS disorder are would be helpful, including what neurodevelopmental difficulties or symptoms may differentiate them. Related to this, there are no p values in table 1, making it difficult to see if there are pre-existing differences between groups. Regarding the EEG/ERP analyses, it is curious to me that the authors had no pre-existing hypotheses concerning the ERPs. The analysis strategy was to "... identifying the prominent peaks in the waveform, extracting the peak amplitude and latency using ERPLab." This seems somewhat non-standard, especially given the plethora of research using the ANT, including in children, that have identified N2 and P3 related findings.It should also be mentioned that relying on peak amplitude (rather than mean amplitude) raises the risk of noise contaminating the findings. This is especially relevant since the sample size is small, which leads to increased noise.
Further, the waveform in the 200-300 ms window appears negative-going (presumably an N2). Why was the positive peak chosen for analysis? It is possible that the window falls into the positive peak, but the figure is hard to read--could a small box be placed around the 200-300 window to make this clearer?
Related to this, the authors report extracting latency but report no latency-related findings. This seems odd given the apparent latency differences between groups in the waveform as it approaches 400 ms, and raises questions related to the authors reporting of amplitude differences in the 200-300 ms window, which may be driven by latency differences.
Also, the authors present baseline ERP waveforms but do not present post-training ERP waveforms. I would like to see the post-training waveforms.
Minor comments: For purposes of clarity, I would like to see an illustration or figure of the task. Because the sample size is so small and there was not an equivalent control (the control group was passive) this study seems very preliminary. Could the word "preliminary" be added to the title somewhere?
Reviewer 2 Report
Summary:
The present paper investigates how short-term musical training affects performance and electrophysiological parameters in an attention task in children with fetal alcohol spectrum disorder. Regarding behavior, a strong trend effect was found for reaction times, demonstrating that the musical training group shows a tendency to react faster in the attention task. For EEG parameters, the training group showed an increased P250 amplitude after training, as well as an increase in post stimulus theta power. The researchers discuss the results in the light of previous studies on musical training, as well as regarding the therapeutical benefits of such a training.
In my opinion, the authors address a highly important topic (i.e., therapeutical intervention for fetal alcohol spectrum disorder) and implement a training and testing-task that is well suited for their sample. The authors further take great care to ensure comparability between the study groups. However, the selection of the control condition is suboptimal, since the control condition consists of a discussion group, which is very passive compared to an active and instructed training. Thus, all results might be potentially unspecific / unrelated to musical training and only due to personal trainer – child interaction. Further, the authors must provide more information on the methods part of the manuscript, in order to ensure that the study can be replicated.
Comments to the Authors:
Abstract:
P: 1, l: 15 Please specify more precisely what is meant with “differences in the brain”.
P: 1, l: 21 Please explicitly state that the test task was performed before and after training. This is implicitly mentioned in the “eight weeks apart”, but this might be unclear for the naïve reader.
P: 1, l: 23 Please also mention the channel label / position here.
P: 1, l: 24 It is unclear to what the term “congruent stimuli” refers to.
P: 1, ll: 27-28 The statement implies a link between the P250 and theta effects and the cognitive concept of attention. This must be better explained/introduced. Currently, the only link provided between the results and the label attention is that the test is called a attention test.
Introduction
The introduction of 1) the task, 2) the current concept of attention, and 3) the neural corelates is much to unspecific. The authors need to provide a concise explanation of 1) how they define attention in this study, 2) how the used test assesses this, and 3) how neural correlates are related to this.
P: 2, l: 46 “including throughout childhood” – This does not seem grammatically correct.
P: 2 ll: 46-48 Compared to whom/which condition did the children with MT show cortical changes?
P: 2, l: 56 Please more precisely link the ANT to the concept of attention. How is attention tested here and which subtype of attention is assessed?
P: 2, ll: 62-63 “improved neural synchrony” - The mechanistic description of the proposed effect is much to coarse. What does improved neural synchrony mean (i.e., Which neurons are synchronous to what? Synchronous at which frequency? From where do we know that improved synchrony is beneficial to performance?)?
Methods:
P: 2, ll: 76-77 Were there any specific exclusion criteria for the EEG measurement?
P: 2, l: 78 How was the random selection performed?
P: 2, l: 87 Please shortly explain how the Kaufman Brief Intelligence Test works for the naïve reader.
P: 3, ll: 109-110 Maybe I missed it, but I do not find a precise statement when relative to the musical training the ANT and EEG measurement was performed. Please specify.
P: 3, ll: 112-113 Please specify how the participants were involved in this. To the naive reader, it seems the participants were not involved here. If this is the case, then an important difference between the experimental and control condition would be personal interaction between a trainer/therapist and the participant, which would introduce a strong confound to the comparison music training vs. control. It would be very different then to interpret any difference between group as an effect of musical training alone.
P: 3, ll: 115-129 Please state the average number of performed trials per child for the ANT task.
P: 3, l: 134 Since no notch/band-stop filter was used, please mention how the 50Hz cutoff here relates to the line noise frequency (i.e., at which frequency was the line noise?)
P: 3, l: 135 Here, it is unclear if “stimulus” refers to the cue or the target stimulus. Please specify.
P: 3, ll: 131-142 Did the researchers apply a (prestimulus) baseline correction, a linear demeaning, or a linear detrending ? If not, please ensure that there are no linear changes in amplitude across trials over the course of the experiment (e.g., due to measurement (electrode impedance changes) or participant-related (e.g., fatigue) reasons)
P: 3, l: 136 Please specify which electrodes were included in the common average reference. Were bad/noisy channels included or rejected?
P: 3, l: 138 Please mention the location of the accessory channels.
P: 4, l: 139 This step is unclear to me. Were all epoched trials for a certain condition averaged?
P: 4, ll: 139-142 Please specify how the peaks were extracted? Which parameter (e.g., maximum absolute peak amplitude) was used for selection? Was there a minimum peak height?
P: 4, l: 144 Please specify what the wavelet parameters refer to. Currently, this is unclear to the naive reader.
P:4, l: 146 Here, the time window is placed both before and after the stimulus. The oscillatory activity before stimulus presentation can hardly be informative of stimulus processing, whereas the oscillatory activity after the stimulus cannot be informative about prestimulus brain states.
Please explain the rationale for the use of the time window. Wouldn't it be more informative to look into prestimulus and poststimulus activity separately?
P: 4, l: 149 This is unclear to me. First, the time window for analysis is stated as 1 s, then only values from a time period of 200 ms are taken into account.
Please explain how this works. If the researchers performed a moving window analysis approach to compute time-resolved theta power across the 1s period, and then selected only poststimulus samples, this needs to be clearly explained, along with all analysis parameters (e.g., step size for window movement).
If this is the case, then the chosen parameters hold a high risk of results being contaminated by prestimulus power differences, which cannot safely be linked to stimulus processing.
P: 4, l: 152 To what does pre/post refer to here? Pre-/Post training? Pre-/Post-stimulus presentation? Please specify.
P: 4, l: 154 Number of multiple comparisons due to multiple electrodes? Why then not average across multiple frontal and central electrodes? Please explain what is meant here.
Results:
P: 4, l: 161 Please explicitely state p-values for each compared metric. It would be a good idea to add this information for all comparisons in table 1.
P: 4, ll: 158-166 What about the behavioral and EEG results for the incongruent and solo conditions? This could be interesting as a baseline/ comparison condition.
P: 5, l: 172 How was the time window of 200-300 ms defined (especially since for the theta power comparison a different time window was selected)? Did the researchers perform multiple comparison correction across time windows / number of compared peaks?
Figure 1A: Would it be possible to likewise plot the waveform for the post-training condition here? E.g., in dotted lines, to allow an easy comparison.
Further please highlight the 200-300 ms window used for comparison (e.g., with grey transparent shading).
Discussion:
P: 6, l: 199 Again, I am concerned about the choice of control condition here. It might very well be that we see an unspecific effect here, with not musical training, but plain trainer-child interaction making the difference here. Please comment on this potential confound. I appreciate the paragraph dealing with this in the limitations section of the discussion, but the point is so important that it needs to addressed more precisely.
Round 2
Reviewer 2 Report
The authors have addressed most of my initial concerns and I think the quality of the manuscript has substantially improved. Importantly, the authors now more specifically explain the concept of attention and its implementation, along with the providing more specific information on the applied methods. Finally, the authors address the issue with the unspecific control group, which provides important recommendations for future (non-pilot) studies.
In the current version of the manuscript, I found a few minor language and labeling inconsistencies. Once they are corrected, I think the manuscript will be ready for publication.
For future studies, I recommend the authors to apply an analysis of spectral data that corrects across multiple comparisons across time (e.g., via cluster-correction), instead of a priori defining a specific analysis window. This approach provides a more objective, precise, and valid assessment of temporal effects.
Minor comments:
Abstract:
P: 1, l: 21 “Selective attention”
P: 1, ll: 28-30 Isn’t this sentence redundant to the previous sentence? I.e., isn’t this just a deconstructed version of the interaction effect? If yes, then this sentence should be removed to avoid confusion.
Introduction:
P: 2, l: 86 Please include a short explanation/practical implementation for the concept of selective attention. Which functional aspect is determined by this?
Methods:
P: 6, ll: 235 Please specify the baseline time window used (i.e., -500 to 0 ms)
P: 7, l: 265 Doesn’t this refer to table 3 instead of to table 2?
P: 7, l: 269 Doesn’t this refer to table 4 instead of to table 3?
Author Response
Responses to Reviewer 2, Revision 2
The authors have addressed most of my initial concerns and I think the quality of the manuscript has substantially improved. Importantly, the authors now more specifically explain the concept of attention and its implementation, along with the providing more specific information on the applied methods. Finally, the authors address the issue with the unspecific control group, which provides important recommendations for future (non-pilot) studies.
In the current version of the manuscript, I found a few minor language and labeling inconsistencies. Once they are corrected, I think the manuscript will be ready for publication.
Response: We thank the author for their careful review of the revised manuscript. We have edited the manuscript to address the remaining minor edits suggested by the reviewer and once again thank the reviewer for their previous helpful comments in improving the manuscript.
For future studies, I recommend the authors to apply an analysis of spectral data that corrects across multiple comparisons across time (e.g., via cluster-correction), instead of a priori defining a specific analysis window. This approach provides a more objective, precise, and valid assessment of temporal effects.
Response: We agree with the reviewer that the best approach is to perform cluster-correction for time/frequency maps. In our understanding the cluster-correction approach performs best with larger sample size datasets, similar to our previous publications that utilized the cluster-correction approach (Stephen et al. 2013, Bolaños et al. 2018). We hope to have a larger dataset with which to implement this more rigorous approach soon.
Minor comments:
Abstract:
P: 1, l: 21 “Selective attention”
Response: We have changed “Attention” to “Selective attention”
P: 1, ll: 28-30 Isn’t this sentence redundant to the previous sentence? I.e., isn’t this just a deconstructed version of the interaction effect? If yes, then this sentence should be removed to avoid confusion.
Response: Yes, the reviewer is correct that this was essentially a restatement of the interaction effect. We have removed the second sentence as suggested by the reviewer to avoid confusion.
Introduction:
P: 2, l: 86 Please include a short explanation/practical implementation for the concept of selective attention. Which functional aspect is determined by this?
Response: We have added a sentence and a reference to better clarify the concept of selective attention and its role in academic performance.
Methods:
P: 6, ll: 235 Please specify the baseline time window used (i.e., -500 to 0 ms)
Response: We added the baseline time interval to this sentence.
P: 7, l: 265 Doesn’t this refer to table 3 instead of to table 2?
Response: Thank you for catching this. We updated the referenced text to Table 3.
P: 7, l: 269 Doesn’t this refer to table 4 instead of to table 3?
Response: Yes, we updated the referenced text to Table 4.